# Isolation and optimization of extracellular PHB depolymerase producer *Aeromonas caviae* Kuk1-(34) for sustainable solid waste management of biodegradable polymers

**Mohammad Amir[1], Naushin Bano[1], Abu Baker[1], Qamar Zia[2,3], Saeed Banawas[2,3,4], Mohd. Rehan Zaheer[5], Mohammad Shariq[6], Md Sarfaraz Nawaz[7], Mohd. Farhan Khan[5,8], Z. R. Azaz Ahmad Azad[9], Anamika Gupta[10], Danish Iqbal [3], Roohi [1]***

1 Protein Research Laboratory, Department of Bioengineering, Integral University, Lucknow, India, 2 Health and Basic Science Research Centre, Majmaah University, Majmaah, Saudi Arabia, 3 Department of Medical Laboratory Sciences, College of Applied Medical Sciences, Majmaah University, Majmaah, Saudi Arabia, 4 Department of Biomedical Sciences, Oregon State University, Corvallis, Oregon, United States of America, 5 Department of Science, Gagan College of Management and Technology, Aligarh, India, 6 Department of Physics, Faculty of Science, Jazan University, Jazan, Saudi Arabia, 7 Department of Chemistry, Faculty of Science, Jazan University, Jazan, Saudi Arabia, 8 Nano Solver Lab, Department of Mechanical Engineering, Z. H. College of Engineering & Technology, Aligarh Muslim University, Aligarh, India, 9 Department of Post-Harvest Engineering and Technology, Aligarh Muslim University, Aligarh, India, 10 Department of Chemistry, Aligarh Muslim University, Aligarh, India

* roohi0607@gmail.com

**Data Availability Statement:** All relevant data are within the paper and its Supporting information

## Abstract

Bioplastics, synthesized by several microbes, accumulates inside cells under stress conditions as a storage material. Several microbial enzymes play a crucial role in their degradation. This research was carried to test the biodegradability of poly-β-hydroxybutyrate (PHB) utilizing PHB depolymerase, produced by bacteria isolated from sewage waste soil samples. Potent PHB degrader was screened based on the highest zone of hydrolysis followed by PHB depolymerase activity. Soil burial method was employed to check their degradation ability at different incubation periods of 15, 30, and 45 days at 37±2˚C, pH 7.0 at 60% moisture with 1% microbial inoculum of *Aeromonas caviae* Kuk1-(34) (MN414252). Without optimized conditions, 85.76% of the total weight of the PHB film was degraded after 45 days. This degradation was confirmed with Fourier-transform infrared spectroscopy (FTIR) and Scanning electron microscope (SEM) analysis. The presence of bacterial colonies on the surface of the degraded film, along with crest, holes, surface erosion, and roughness, were visible. Media optimization was carried out in statistical mode using Plackett Burman (PB) and Central Composite Design (CCD) of Response Surface Methodology (RSM) by considering ten different factors. Analysis of Variance (ANOVA), Pareto chart, response surface plots, and F-value of 3.82 implies that the above statistical model was significant. The best production of PHB depolymerase enzyme (14.98 U/mL) was observed when strain Kuk1-(34) was grown in a media containing 0.1% PHB, $K_2HPO_4$ (1.6 gm/L) at 27 °C for seven days. Exploiting these statistically optimized conditions, the culture was found to be a suitable candidate for the

files, except some files that are available from the NCBI database (accession numbers MN088848, MN736124, MN414252)

**Funding:** RR is grateful to the Department of Science and Technology, India (DST-SERB) for providing financial support under the scheme of Early Career Research Award (Project No. ECR/ 2017/001001). SB would like to thank Deanship of Scientific Research at Majmaah University for supporting this work under Project Number No. R- 2022-58. The funders had no role in study design, data collection and analysis, decision to publish, or preparation of the manuscript.

**Competing interests:** The authors have declared that no competing interests exist.

management of solid waste, where 94.4% of the total weight of the PHB film was degraded after 45 days of incubation.

## 1. Introduction

Plastics play an essential role in daily life; hence, their requirement increased tremendously from 1.5 million tonnes in 1950 [1] to ~300 million tons in 2015 [2]. These synthetic polymers are usually low-cost but have a major negative impact on our environment [3]. Due to their non-degradable nature, plastics are the main culprit of environmental nuisance. Despite many recycling efforts, the results of plastics disposal in municipal landfills still create significant problems. Consequently, many attempts have been undertaken to generate renewable, degradable, and recyclable materials, i.e., green materials, for sustainability [4, 5]. Poly(lactic acid) (PLA), poly(butylene succinate adipate) (PBSA), polycaprolactone (PCL), and poly(hydroxy alkanoates) (PHAs) are few biodegradable aliphatic polymers, which can replace polyethylene (PE) and polystyrene (PS) that took hundreds or thousands of years to degrade [4].

PHAs are produced by a diversity of microorganisms as carbon and energy storage material under stress conditions. Poly-β-hydroxybutyrate (PHB) is the most commonly occurring PHA, which comprises of packed monomers of (R)-3-hydroxybutyric acid (R3HB) [6, 7]. PHB depolymerase (EC 3.1.1.75) is an extracellular and intracellular hydrolyzing enzyme that degrades PHB effectively [8]. Partially structured (denatured) PHB is degraded by extracellular depolymerase [9], while intracellular depolymerase operates on unstructured (native) PHB [10]. A plethora of reports are available on the biodegradation of PHAs in a terrestrial, marine, soil environment [11–13]. As a result, several microbes have been characterized responsible for the degradation of these polymers [14]. In the present study, biodegradation of PHB polymer from a novel bacterial enzyme isolated from the sewage waste soil bacteria was observed. Their detailed statistical production optimizations were also studied, including soil burial applications for the sustainability of the environment.

## 2. Materials and methods

### 2.1. Polymer studied

PHB powder with a linear formula of $[COCH_2CH(CH_3)O]_n$ was purchased from Sigma-Aldrich, CAS Number: 29435-48-1.

### 2.2. Sample collection and isolation of PHB degrading microbes

Soil samples from sewage sludge in semi-solid form were taken from four different sources in Lucknow, Uttar Pradesh, India (Table 1) for isolation of potent PHB depolymerase-producing microorganisms. Samples were processed by serial dilution following spread plate method on

Table 1. Soil samples taken from different locations of Lucknow, UP, India.

| Sampling site | Latitude/Longitude | Total Isolates | Positive Isolates |
|---|---|---|---|
| Kukrail | 26.91 N/ 80.98 E | 73 | 15 |
| Gomti Nagar | 26.84 N/ 81.00 E | 31 | 04 |
| IIM Road | 26.80 N/ 80.76 E | 16 | 02 |
| Molviganj | 26.85 N/ 80.92 E | 07 | 01 |

Bushnell Hass medium and incubated for 48 h at 37 ºC for the proliferation of microbial growth and isolation of microorganisms.

## 2.3. Screening of PHB depolymerase producing isolates

PHB depolymerase producers were screen out by clear zone assay on Bushnell Hass Medium (BHM) (g/L): Magnesium sulphate, 0.20 g; Calcium chloride anhydrous, 0.02 g; Potassium di-hydrogen phosphate, 1.0 g; Di-potassium hydrogen phosphate, 1.0 g; Ammonium nitrate, 1.0 g; Ferric chloride, 0.05 g with 0.15% PHB powder sonicated in an ultrasonic water bath (Labman-LMUC 3, 40 KHz and 100 W) at 40 ºC for 20 min followed by addition of 2% agar powder to generate solid media. After sonication, plates were incubated for seven days at 37 ºC, and then a clear hydrolysis zone was measured.

## 2.4. PHB depolymerase assay

PHB depolymerase assay was performed as per the modified method of Kobayashi et al. [15]. Tris-HCL buffer (50 mM, pH 7.0) with 0.15% of PHB powder was suspended and subjected to sonication immersed in an ultrasonic water bath (40 kHz and 100 W) for 30 min. In 0.9 mL of the substrate suspension, 0.1 mL of culture supernatant was added and incubated for 25 min at 37 ºC. The activity of the decrease in turbidity of the PHB suspension was measured at $OD_{650}$ against blank (1 mL of Tris HCL buffer) [14].

## 2.5. Morphological, physiological, biochemical identification of isolates

The positive bacterial isolates that gave a clear zone of hydrolysis on PHB plates were identified in accordance with Bergey's Manual [16]. Detailed analysis was done for morphological, physiological, and biochemical identification. The 16S rRNA gene sequence analysis was performed by Biokart, Bangalore, India Pvt. Ltd. using forward primer sequence 27F and backward primer sequence 149R [17]. The sequences were examined and compared to the nucleotide sequences stored in the NCBI (National Center for Biotechnology Information) database using Basic Local Alignment Search Tool (BLAST) search engine. The phylogenetic tree, as implemented in MEGA X from nucleotide sequences, was developed using the neighbouring ClustalW method [18].

## 2.6. Preparation of PHB film

PHB film was prepared by solvent casting method where 0.1 g of PHB powder (Sigma-Aldrich) was suspended in 30 mL of chloroform while kept on a magnetic stirrer for 20 min at 45 ºC. PHB-suspension was poured into clean autoclaved glass Petri plates, and chloroform was then vaporized. Petri plates were further incubated at 37 ºC for 24 h, resulting in the formation of PHB films about 10 cm in diameter and 2 mm in thickness [19].

## 2.7. Soil burial biodegradation analysis with PHB film

The change in the weight of PHB polymer film was calculated before and after the treatment with the positive bacterial strains. PHB films (0.024–0.026 g) were buried in autoclaved soil in pots (at 3 cm depth) at ambient temperature (35–37˚C) with moisture content uphold to 60% in the presence of mineral salts and incubated for specified time period (i.e. 15, 30 and 45 days). The pre-weighed PHB polymer films were then taken off from the soil after different incubation times and washed several times by distilled water to remove soil particles and then

dried at room temperature [17]. The polymer degradation was calculated by given formula:

$$\% \text{ Degradation} = \left[ \frac{(W_{if} - W_{ff})}{W_{if}} \right] \times 100$$

where, $W_{if}$ = initial weight of films, $W_{ff}$ = final weight of buried films.

## 2.8. Scanning Electron Microscopy (SEM) analysis of soil buried PHB films

Soil fragmented PHB film surface was analyzed using SEM for detailed visualization [17]. Before SEM analysis, the PHB films were fixed in 4% (v/v) glutaraldehyde in sodium cacodylate buffer (100 mM, pH 7.2). After the glutaraldehyde fixation, PHB films were again fixed in 20 g/L of aqueous osmium tetroxide and, after drying to the critical point, examined under SEM [20].

## 2.9. Fourier-Transform Infra-Red (FTIR) spectroscopic analysis

FTIR spectra of PHB polymer film were obtained using a Perkin Elmer System 2000 Fourier transform infrared spectrometer. A small amount of PHB polymer film was immersed in an organic solvent, milled with KBr, and pressed into a transparent film for FTIR analysis. FTIR spectra were collected over the range of 4000 to 450 cm$^{-1}$ [21]. FTIR determined functional groups in the degradation patterns were compared with control [22].

## 2.10. Statistical optimization of the media variables for PHB depolymerase enzyme production using Placket-Burman (PB) experimental design

PB experimental design was used in the initial phase of optimization using Minitab 19. In this experiment, ten independent variables were used, and each variable was checked at high (+1) and low (-1) values. All experiments were carried out in triplicates and repeated twice. PHB depolymerase activity was used as a response based on experimental design and polynomial model of the first order as follows:

$$Y = \beta_0 \cdot \Sigma \, \beta_i \, X_i$$

where, Y is response (enzyme production), $\beta_0$ is model intercepts, $\beta_1$ is a linear coefficient, and $X_i$ is the level of the independent variable. This model was used to screen and assess the key factors influencing response. The *p* value ≤0.05 was a probability that defines the magnitude of a contrast coefficient arising from the variability of the random process and was calculated using Analysis of Variance (ANOVA). A Pareto chart analysis was drawn using standardized effects, and the results of F-value validate the importance of significant effects [23].

## 2.11. Statistical optimization of the PB variables for PHB depolymerase enzyme production using Response Surface Methodology (RSM)

Response surface central composite design (CCD) specifies optimal concentration of the significant variables and the interacting effect of media ingredients obtained in PB design. Out of these ten variables, five of them (namely time, temperature, PHB, $KH_2PO_4$, and $K_2HPO_4$) were further optimized with 32 runs of experiments, while the rest five variables remained constant. A high (+2) to low (-2) values have been checked for each variable. The model was validated by ANOVA and response surface plots to ensure efficiency.

## 2.12. Soil bioremediation and solid waste management under statistically optimized conditions

The biodegradation of PHB-based biofilm treated with bacterial strain by soil burial method under statistically optimized conditions for the application of solid waste management was performed as previously described in section 2.7 for 15, 30, and 45 days and the polymer degradation was calculated in the similar manner [22].

## 3. Results

### 3.1. Screening of PHB depolymerase isolates

In the present study, a total of four sewage waste soil samples were collected from different dumping zones of Lucknow. These samples produced maximum number of bacterial strains that were found to be potent producers of PHB depolymerase and were capable of degrading PHB-based bioplastics. After the serial dilution, the sewage waste soil samples were spread on BHM+PHB agar plates. Among 127 isolates, a total of 22 positive PHB degraders were selected based on their ability to form a zone of hydrolysis on PHB agar plate (Fig 1A and 1B). Out of these 22 bacterial strains, three were selected on the basis of large zone of diameter, and named as CB2-(20), Kuk1-(34) & CA6-(55). Zone diameters were observed in the range of 2.5 to 11.3 mm (Table 2). Zone of hydrolysis showing enzyme production is clearly visible in the Fig 1C–1E.

### 3.2. PHB depolymerase activity

The degradation ability of 22 positive isolates was also checked by PHB depolymerase assay. The isolates, CB2-(20), Kuk1-(34) & CA6-(55), exhibited potent activity in the range of 1.55–2.06 U/mL/min (Table 2). Among these three significant PHB degraders, Kuk1-(34) gave the

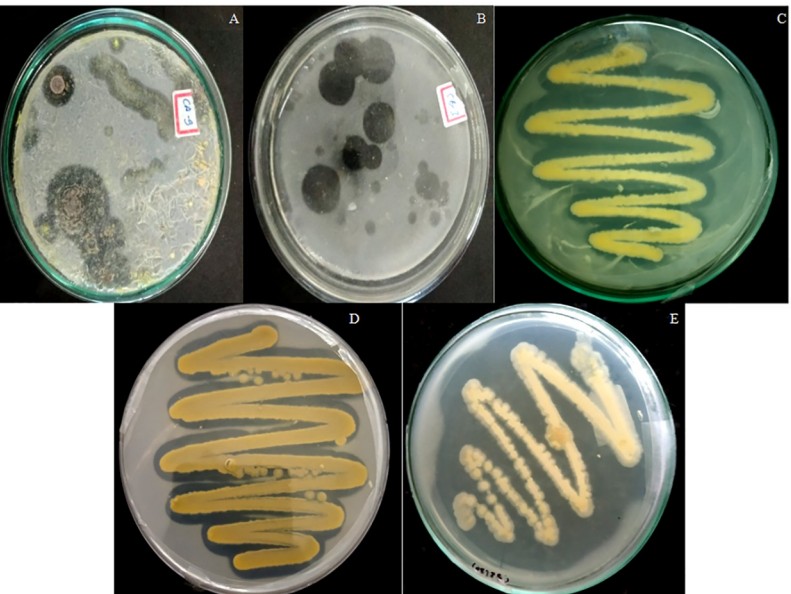

**Fig 1.** Images showing clear zone of diameter of microbial colonies in (A) and (B). Positive colonies showing PHB hydrolysis on PHB agar media (C) CA6-(55), (D) CB2-(20), and (E) Kuk1-(34).

**Table 2. Enzyme activity and CFU count of positive bacterial isolates.**

| S. No | Positive isolates | Colony size in mm (a) | Zone of hydrolysis in mm (b) | Zone diameter in mm (b/a) | Enzyme activity (U/mL/min) | Decrease in turbidity measured at $OD_{650}$ | CFU/mL |
|---|---|---|---|---|---|---|---|
| 1 | Control | NA | NA | NA | NA | 0.823 | NA |
| 2 | CA6-(55) | 4 | 10 | 2.5 | 1.55 | 0.379 | $1.29 \times 10^9$ |
| 3 | CB2-(20) | 4 | 24 | 6 | 1.71 | 0.404 | $4.80 \times 10^9$ |
| 4 | Kuk1-(34) | 3 | 34 | 11.3 | 2.06 | 0.320 | $4.10 \times 10^9$ |

best PHB depolymerase activity (2.06 U/mL/min) and maximum zone of hydrolysis (11.3 mm) and, therefore, was selected for further study.

## 3.3. Morphological, biochemical, physiological, and phylogenetic analysis

All the physical and physiological characteristics of bacterial strain Kuk1-(34) are represented in Table 3. Phylogenetic evaluation of the Kuk1-(34) strain was concluded with the help of alignment and cladistics analysis of a homologous sequence of known bacteria. 16S rRNA sequences were submitted in NCBI through GenBank(accession number MN414252), and their identity was performed through BLAST. The phylogenetic tree of the submitted strain in the NCBI is shown in Fig 2, and the strain was identified as *Aeromonas caviae* Kuk1-(34) sp (Table 4).

**Table 3. Morphological, biochemical and physiological characteristics of the bacterial strain Kuk1-(34).**

| Characteristics | Bacterial strain Kuk1-(34) |
|---|---|
| **Morphological tests** | |
| Grams staining | Negative |
| Pigmentation | Yellow |
| Form | Irregular |
| Elevation | Umbonate |
| Cell shape | Rod |
| Margin | Undulate |
| **Biochemical tests** | |
| Cellulose | + |
| Casein | + |
| Indole | - |
| Methyl red | - |
| Voges-Proskauer | - |
| Citrate | + |
| Hydrogen sulphide | + |
| Catalase | + |
| **Physiological tests** | |
| Optimum temperature for growth | 17–37 °C |
| Growth at NaCl (%) | 0.5–5 |
| Optimum pH for growth | 6.0–8.0 |

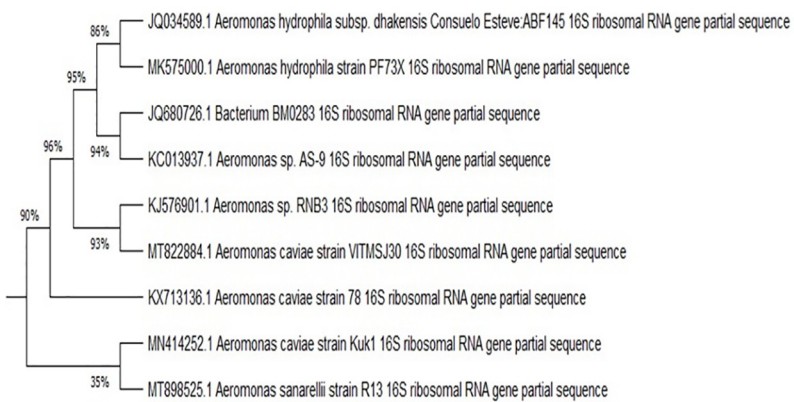

**Fig 2. Phylogenetic tree of *Aeromonas caviae* Kuk1-(34) sp. drawn using UPGMA tree (MEGA X software) with the evolutionary distances showing the relationship of PHB depolymerase producing bacteria with the known sequences of related genera.**

## 3.4. Weight loss analysis of PHB film by soil burial method

PHB films (0.024–0.026 gm) were buried in 200 g of sterile soil inside the pot with different strains, viz. Kuk1-(34), CA6-(55) and CB2-(20), separately. It was observed that the weight of PHB film was maximally reduced 45 days after pre-treatment with strain *A. caviae* Kuk1-(34) sp. as compared to the other isolates (Fig 3). Up to 85.76% degradation of PHB film was observed in comparison to the control (Table 5) under unoptimized conditions.

## 3.5. SEM analysis of soil buried PHB films

In the present study, soil-buried PHB films of 15, 30, and 45 days were analyzed by SEM. Several holes, crests, surface erosion, and significant roughness were clearly observed on all PHB films as compared to the control. Multiple bacterial colonies of *A.caviae* Kuk1-(34) sp. were seen attached on the surface of the PHB films, which implies that this species actually secretes extracellular depolymerase enzyme that degrades PHB film (Fig 4).

## 3.6. FTIR Analysis of soil buried PHB films

FTIR analysis of soil buried PHB film treated with Kuk1-(34) sp. showed changes in the functional groups and significant shift of wave-numbers, indicating biodegradation of PHB film. It was observed that in control, the FTIR chromatogram peak for functional group O-H appearing at 3435.25 cm$^{-1}$ shifted to 3392.51 cm$^{-1}$ after 15 days incubation with Kuk1-(34) strain. This peak moved further to 3401.97 cm$^{-1}$ and 3409.23 cm$^{-1}$ with an intensity of 62.2 and 44.67, up on increasing the incubation time to 30 and 45 days, respectively. The functional group, ester bond, present in control showed a characteristic peak at 1729.63 cm$^{-1}$; while in Kuk1-(34) treated samples, this peak shifted to 1623.16, 1724.7 and 1633.04 cm$^{-1}$ respectively, after

**Table 4. 16S rRNA identification of positive isolates.**

| S.No. | Submitted microbial strain | Identity | Accession Number | Similarity Index % |
|---|---|---|---|---|
| 1. | CA6-(55) | *Enterobacter cloacae* CA655 | MN088848 | 97.88 |
| 2. | CB2-(20) | *Stenotrophomonas* sp. CB220 | MN736124 | 96.60 |
| 3. | Kuk1-(34) | *Aeromonas caviae* Kuk1 sp. | MN414252 | 99.43 |

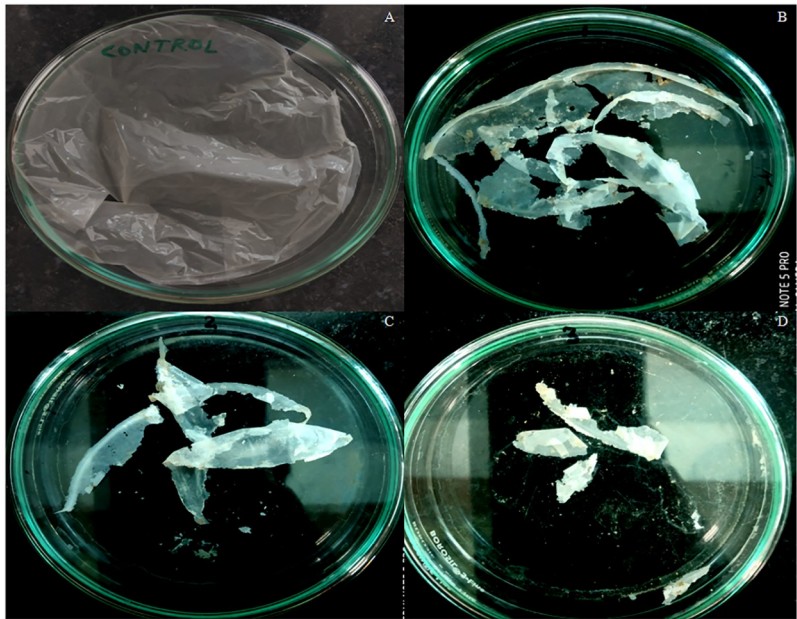

**Fig 3. Degradation of PHB film treated with (A) control and *Aeromonas caviae* Kuk1-(34) sp. after (B) 15 days (C) 30 days and (D) 45 daysin soil burial method.**

incubation for 15, 30, and 45 days. There is the formation of a new-fangled bond and an additional functional group after degradation at 45 days, emerging as a peak at 2923.08 cm$^{-1}$ with an intensity of 47.84. This supplementary peak could be due to the -C-H functional group. Another peak at 1047.18 cm$^{-1}$ with an intensity of 46.02 is indicative of the -C-O functional group. These changes in wavelength frequency of major functional groups in degrading samples as compared to control give preliminary proof of biodegradation of PHB film by *A. caviae* Kuk1-(34) sp. after incubation for 45 days (Fig 5).

## 3.7. Statistical optimization to determine the interactive impact of Kuk1-(34) sp. on enzyme production using PB design

For PB analysis, ten independent variables were selected (Table 6). The experiment was carried out with 12 runs in different combinations to identify significant parameters for extracellular PHB depolymerase production. Higher (+1) and lower (-1) values were screened for each variable (Table 7). Experimental data was statistically analyzed using F-test for ANOVA. The model F-value of 202.74 implies that the model was significant, and $p$-values were used as a tool to check the significance of each parameter (Table 8). Thus, $p$-values $<0.05$ denoted the importance of factors on enzyme production. The Pareto Chart illustrates the order of

**Table 5. Weight loss degradability of PHB film after different incubation time in un-optimised conditions.**

| Bacterial species | Weight of PHB film (g) | Weight of pre-treated soil buried PHB film (g) | | | Weight loss degradability |
|---|---|---|---|---|---|
| | | 15 days | 30 days | 45 days | After 45 days (%) |
| Control | 0.0240 | 0.0240 | 0.0240 | 0.0240 | 0.00% |
| CA6-(55) | 0.0250 | 0.0224 | 0.0190 | 0.0145 | 42.00% |
| CB2-(20) | 0.0240 | 0.0229 | 0.0202 | 0.0170 | 29.16% |
| Kuk1-(34) | 0.0260 | 0.0232 | 0.0187 | 0.0037 | 85.76% |

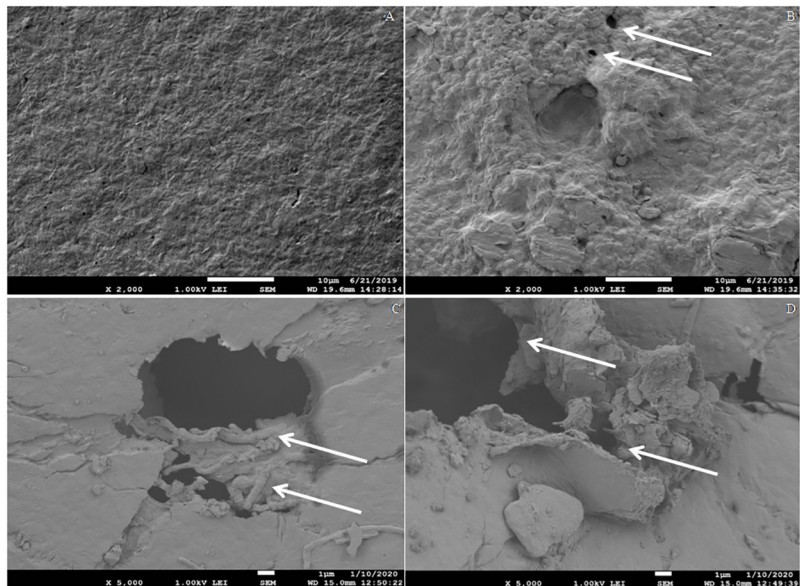

**Fig 4. Morphological changes in SEM micrographs of PHB film degraded in soil burial method after incubation with (A) Control PHB film without treatment, and *Aeromonas caviae* Kuk1-(34) sp after (B) 15 days (C) 30 days (D) 45 days.**

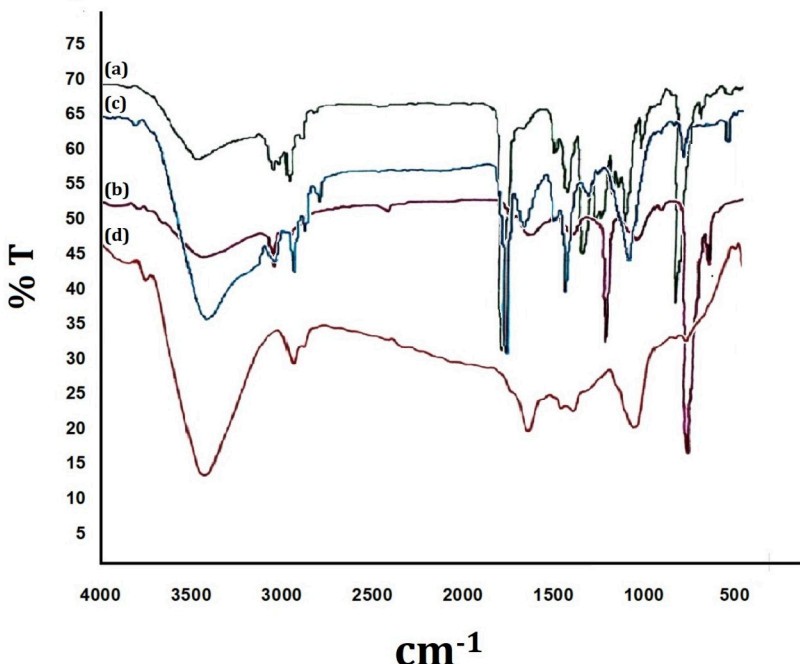

**Fig 5. FTIR chromatogram of soil buried PHB film after the degradation by *Aeromonas caviae* Kuk1-(34) sp.** The functional groups changes were compared with each film after incubation with (A) Control PHB film without treatment, *Aeromonas caviae* Kuk1-(34) sp after (B) 15 days (C) 30 days (D) 45 days.

**Table 6. Component design of Plackett Burman for PHB depolymerase production with the bacterial strain *Aeromonas caviae* Kuk1-(34) sp.**

| Variable | Symbol | (+1) High value | (0) Central value | (-1) Low Value |
|---|---|---|---|---|
| Time (days) | A | 7 | 5 | 3 |
| Temp (ºC) | B | 47 | 37 | 27 |
| pH | C | 9 | 7 | 5 |
| PHB (%) | D | 0.2 | 0.15 | 0.1 |
| $MgSO_4$ (g/L) | E | 0.25 | 0.20 | 0.15 |
| $CaCl_2$ (g/L) | F | 0.005 | 0.003 | 0.001 |
| $KH_2PO_4$ (g/L) | G | 1.3 | 1 | 0.7 |
| $K_2HPO_4$ (g/L) | H | 1.3 | 1 | 0.7 |
| $NH_4NO_3$ (g/L) | I | 1.3 | 1 | 0.7 |
| $FeCl_3$ (g/L) | J | 0.07 | 0.05 | 0.03 |

**Table 7. Placket Burman experimental design for PHB depolymerase production with the bacterial strain *Aeromonas caviae* Kuk1-(34) sp.**

| Run Order | (A) | (B) | (C) | (D) | (E) | (F) | (G) | (H) | (I) | (J) | Enzyme activity (U/mL) |
|---|---|---|---|---|---|---|---|---|---|---|---|
| 1 | -1 | 1 | -1 | -1 | -1 | 1 | 1 | 1 | -1 | 1 | 3.467 |
| 2 | 1 | 1 | -1 | 1 | 1 | -1 | 1 | -1 | -1 | -1 | 0.967 |
| 3 | -1 | -1 | -1 | 1 | 1 | 1 | -1 | 1 | 1 | -1 | 9.8 |
| 4 | -1 | 1 | 1 | 1 | -1 | 1 | 1 | -1 | 1 | -1 | 0.701 |
| 5 | 1 | 1 | 1 | -1 | 1 | 1 | -1 | 1 | -1 | -1 | 1.048 |
| 6 | 1 | -1 | -1 | -1 | 1 | 1 | 1 | -1 | 1 | 1 | 1.88 |
| 7 | 1 | 1 | -1 | 1 | -1 | -1 | -1 | 1 | 1 | 1 | 3.79 |
| 8 | -1 | -1 | 1 | 1 | 1 | -1 | 1 | 1 | -1 | 1 | 9.25 |
| 9 | 1 | -1 | 1 | -1 | -1 | -1 | 1 | 1 | 1 | -1 | 2.9 |
| 10 | 1 | -1 | 1 | 1 | -1 | 1 | -1 | -1 | -1 | 1 | 6.153 |
| 11 | -1 | 1 | 1 | -1 | 1 | -1 | -1 | -1 | 1 | 1 | 0.887 |
| 12 | -1 | -1 | -1 | -1 | -1 | -1 | -1 | -1 | -1 | -1 | 8.38 |

**Table 8. ANOVA analysis for Plackett Burman design for PHB depolymerase enzyme production.**

| Source | DF | Adj SS | Adj MS | F-Value | *p*-Value |
|---|---|---|---|---|---|
| Model | 10 | 129.354 | 12.9354 | 202.74 | 0.055 |
| Linear | 10 | 129.354 | 12.9354 | 202.74 | 0.055 |
| Time (Days) | 1 | 20.664 | 20.6640 | 323.88 | 0.035 |
| Temperature (ºC) | 1 | 63.035 | 63.0346 | 987.97 | 0.020 |
| pH | 1 | 4.496 | 4.4958 | 70.46 | 0.075 |
| PHB (%) | 1 | 12.199 | 12.1988 | 191.20 | 0.046 |
| $MgSO_4$ (g/L) | 1 | 0.203 | 0.2025 | 3.17 | 0.326 |
| $CaCl_2$(g/L) | 1 | 0.814 | 0.8138 | 12.76 | 0.174 |
| $KH_2PO_4$(g/L) | 1 | 9.888 | 9.8881 | 154.98 | 0.051 |
| $K_2HPO_4$(g/L) | 1 | 10.616 | 10.6164 | 166.40 | 0.049 |
| $NH_4NO_3$(g/L) | 1 | 7.218 | 7.2184 | 113.14 | 0.060 |
| $FeCl_3$(g/L) | 1 | 0.222 | 0.2217 | 3.47 | 0.313 |
| Error | 1 | 0.064 | 0.0638 | | |
| Total | 11 | 129.418 | | | |

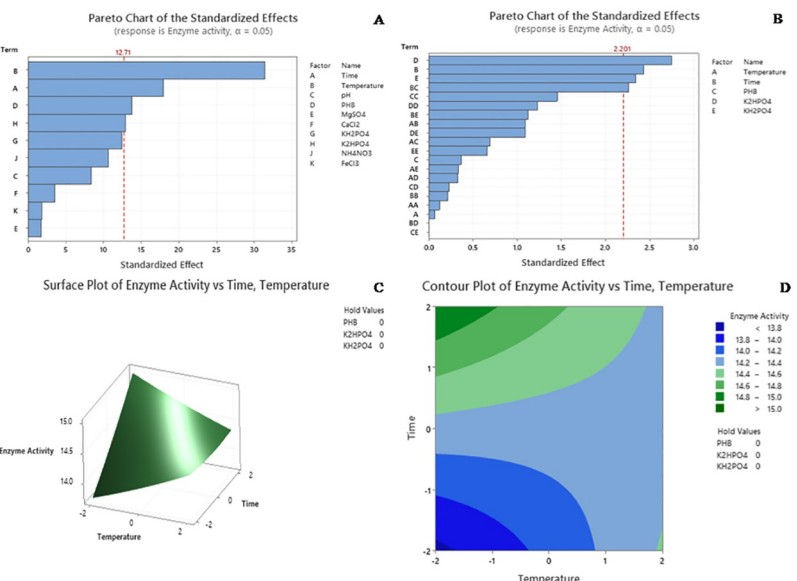

**Fig 6. Pareto chart of the significant factor in the (A) Plackett Burman (PB) design, (B) Central composite design (CCD) for the enzyme production,(C) Surface plot of enzyme activity of the bacterial strain *Aeromonas caviae* Kuk1-(34) sp., (D) Contour plot showing the interaction effect of temperature and time on PHB depolymerase enzyme production by *Aeromonas caviae* Kuk1-(34) sp.**

significance of the variables affecting the Kuk1-(34) production in PB design (Fig 6A). Among ten variables, temperature showed the highest positive effect, followed by time, PHB, and $K_2HPO_4$; while $KH_2PO_4$ comes near the positive and effective parameter. The negative impact was demonstrated by $NH_4NO_3$, pH, $CaCl_2$, $FeCl_3$, and $MgSO_4$. Therefore, for the next optimization step, the optimum level of time, temperature, PHB, $K_2HPO_4$, and $KH_2PO_4$ were checked by RSM using CCD. The model equation for enzyme production (Y) can be written as:

$$Y = 4.109 - 1.3123\,\text{Time} - 2.2919\,\text{Temperature} - 0.6121\,\text{pH} + 1.0083\,\text{PHB}$$
$$- 0.1299\,\text{MgSO}_4 - 0.2604\,\text{CaCl}_2 - 0.9078\,\text{KH}_2\text{PO}_4 - 0.7756\,\text{NH}_4\text{NO}_3 + 0.1359\,\text{FeCl}_3$$

## 3.8. Statistical optimization of significant factors of PB plots by CCD

Significant factors of the PB experiment were then further tested by CCD to evaluate the optimized value for extracellular PHB depolymerase production. Time, temperature, PHB, $K_2HPO_4$, and $KH_2PO_4$ were assessed using Minitab 19 software (Table 9). Each variable was tested from high (+2) to low (–2) levels (Table 10). The ANOVA result showed that the

**Table 9. Component design of CCD for PHB depolymerase production with the bacterial strain *Aeromonas caviae* Kuk1-(34) sp.**

| Variable | Symbol | (+2) value | (+1) value | (0) | (-1) Value | (-2) Value |
|---|---|---|---|---|---|---|
| Time (days) | A | 9 | 7 | 5 | 3 | 1 |
| Temp (ºC) | B | 57 | 47 | 37 | 27 | 17 |
| PHB (%) | D | 0.25 | 0.2 | 0.15 | 0.1 | 0.05 |
| $KH_2PO_4$ (g/L) | G | 1.6 | 1.3 | 1 | 0.7 | 0.4 |
| $K_2HPO_4$ (g/L) | H | 1.6 | 1.3 | 1 | 0.7 | 0.4 |

**Table 10. Central Composite Design for PHB depolymerase production with the bacterial strain *Aeromonas caviae* Kuk1-(34) sp.**

| Run Order | A | B | D | G | H | Enzyme activity (U/mL) | |
|---|---|---|---|---|---|---|---|
| | | | | | | Predicted value | Experimental value |
| 1 | 0 | 0 | 0 | 2 | 0 | 13.87346 | 13.87 |
| 2 | -1 | 1 | -1 | -1 | -1 | 14.14433 | 14.09 |
| 3 | 0 | -2 | 0 | 0 | 0 | 14.32713 | 14.71 |
| 4 | 0 | 0 | 0 | 0 | 0 | 14.30838 | 14.11 |
| 5 | 0 | 2 | 0 | 0 | 0 | 14.34463 | 13.98 |
| 6 | 1 | 1 | 1 | -1 | -1 | 14.29292 | 14.38 |
| 7 | -1 | -1 | 1 | 1 | 1 | 14.23675 | 14.17 |
| 8 | 1 | 1 | -1 | -1 | 1 | 14.60967 | 14.80 |
| 9 | -2 | 0 | 0 | 0 | 0 | 14.05696 | 14.29 |
| 10 | 0 | 0 | 0 | 0 | 0 | 14.30838 | 14.1 |
| 11 | 1 | 1 | -1 | 1 | -1 | 14.11608 | 14.18 |
| 12 | 0 | 0 | 0 | 0 | 0 | 14.30838 | 14.47 |
| 13 | 1 | -1 | -1 | -1 | -1 | 14.60925 | 14.48 |
| 14 | -1 | -1 | -1 | 1 | -1 | 13.40492 | 13.17 |
| 15 | 0 | 0 | 0 | 0 | -2 | 14.24346 | 14.53 |
| 16 | -1 | 1 | 1 | 1 | -1 | 13.99858 | 13.98 |
| 17 | -1 | -1 | -1 | -1 | 1 | 14.2325 | 14.13 |
| 18 | 2 | 0 | 0 | 0 | 0 | 14.65479 | 14.44 |
| 19 | -1 | -1 | 1 | -1 | -1 | 14.29175 | 14.08 |
| 20 | -1 | 1 | -1 | 1 | 1 | 13.95933 | 14.05 |
| 21 | 0 | 0 | 0 | 0 | 0 | 14.30838 | 14.41 |
| 22 | 0 | 0 | 0 | 0 | 0 | 14.30838 | 14.53 |
| 23 | 0 | 0 | 0 | 0 | 2 | 14.91829 | 14.65 |
| 24 | 0 | 0 | 0 | -2 | 0 | 14.44829 | 14.47 |
| 25 | 1 | -1 | 1 | 1 | -1 | 14.0125 | 13.91 |
| 26 | 1 | -1 | 1 | -1 | 1 | 14.43308 | 14.47 |
| 27 | 1 | 1 | 1 | 1 | 1 | 14.37792 | 14.61 |
| 28 | 0 | 0 | 0 | 0 | 0 | 14.30838 | 14.2 |
| 29 | 1 | -1 | -1 | 1 | 1 | 14.96425 | 14.98 |
| 30 | 0 | 0 | -2 | 0 | 0 | 13.93996 | 14.008 |
| 31 | -1 | 1 | 1 | -1 | 1 | 14.75617 | 14.87 |
| 32 | 0 | 0 | 2 | 0 | 0 | 14.02979 | 13.98 |

regression is statistically significant for enzyme production (Table 11). The F-value of 3.82 implies that the model was significant. The Pareto Chart (Fig 6B) illustrates the order of significance of the variables affecting the production of an enzyme from Kuk1-(34) sp. The autonomous $K_2HPO_4$ showed the most significant beneficial impact among all factors, followed by the independent time variable, autonomous $KH_2PO_4$ variable, and finally, interaction between time and PHB that also showed the beneficial impact. Each response surface for enzyme activity indicated a clear peak, which means that the optimum point was inside the design boundary level (Fig 6C). The effect of time and temperature on enzyme activity while keeping PHB, $K_2HPO_4$, and $KH_2PO_4$ at zero level was depicted. The response surface plot showed that the maximum production of an enzyme could be attained at optimum temperature (27 °C) and maximum incubation time (3 days). As a result, enzyme production increased exponentially with a decrease in the incubation time and keeping the temperature at an optimum level. By

**Table 11. ANOVA analysis of CCD for PHB depolymerase enzyme production.**

| Source | DF | Adj SS | Adj MS | F-Value | p-Value |
|---|---|---|---|---|---|
| Model | 20 | 2.99525 | 0.149763 | 1.66 | 0.195 |
| Linear | 5 | 1.72742 | 0.345484 | 3.82 | 0.030 |
| Temperature (°C) | 1 | 0.00046 | 0.000459 | 0.01 | 0.944 |
| Time (Days) | 1 | 0.53611 | 0.536107 | 5.93 | 0.033 |
| PHB (%) | 1 | 0.01211 | 0.012105 | 0.13 | 0.721 |
| $K_2HPO_4$ (g/L) | 1 | 0.68310 | 0.683100 | 7.56 | 0.019 |
| $KH_2PO_4$ (g/L) | 1 | 0.49565 | 0.495650 | 5.49 | 0.039 |
| Square | 5 | 0.40545 | 0.081090 | 0.90 | 0.516 |
| Temperature*Temperature | 1 | 0.00139 | 0.001386 | 0.02 | 0.904 |
| Time*Time | 1 | 0.00414 | 0.004136 | 0.05 | 0.834 |
| PHB*PHB | 1 | 0.19186 | 0.191862 | 2.12 | 0.173 |
| $K_2HPO_4$*$K_2HPO_4$ | 1 | 0.13614 | 0.136136 | 1.51 | 0.245 |
| $KH_2PO_4$*$KH_2PO_4$ | 1 | 0.03989 | 0.039886 | 0.44 | 0.520 |
| 2-Way Interaction | 10 | 0.86238 | 0.086238 | 0.95 | 0.525 |
| Temperature*Time | 1 | 0.10808 | 0.108077 | 1.20 | 0.297 |
| Temperature*PHB | 1 | 0.04337 | 0.043368 | 0.48 | 0.503 |
| Temperature*$K_2HPO_4$ | 1 | 0.00985 | 0.009851 | 0.11 | 0.747 |
| Temperature*$KH_2PO_4$ | 1 | 0.01015 | 0.010151 | 0.11 | 0.744 |
| Time*PHB | 1 | 0.46410 | 0.464102 | 5.14 | 0.045 |
| Time*$K_2HPO_4$ | 1 | 0.00001 | 0.000005 | 0.00 | 0.994 |
| Time*$KH_2PO_4$ | 1 | 0.11408 | 0.114075 | 1.26 | 0.285 |
| PHB*$K_2HPO_4$ | 1 | 0.00501 | 0.005006 | 0.06 | 0.818 |
| PHB*$KH_2PO_4$ | 1 | 0.00000 | 0.000001 | 0.00 | 0.998 |
| $KH_2PO_4$* $KH_2PO_4$ | 1 | 0.10775 | 0.107748 | 1.19 | 0.298 |
| Error | 11 | 0.99380 | 0.090345 | | |
| Lack-of-Fit | 6 | 0.81247 | 0.135412 | 3.73 | 0.085 |
| Pure Error | 5 | 0.18133 | 0.036265 | | |
| Total | 31 | 3.98905 | | | |

applying multiple regression analysis on experimental data, the following second-order polynomial equation was obtained to describe the enzyme production efficiency (Y):

$$Y = 14.308 + 0.0044 \text{ Temperature} + 0.1295 \text{ Time} + 0.0225 \text{ PHB} + 0.1687 \text{ K}_2\text{HPO}_4$$
$$- 0.1437 \text{ KH}_2\text{PO}_4 + 0.0069 \text{ Temperature} * \text{Temperature} + 0.0119 \text{ Time} * \text{Time}$$
$$- 0.0809 \text{ PHB} * \text{PHB} + 0.0681 \text{ K}_2\text{HPO}_4 * \text{K}_2\text{HPO}_4 - 0.0369 \text{ KH}_2\text{PO}_4 * \text{KH}_2\text{PO}_4$$
$$- 0.0822 \text{ Temperature} * \text{Time} + 0.0521 \text{ Temperature} * \text{PHB} - 0.0248 \text{ Temperature}$$
$$* \text{K}_2\text{HPO}_4 - 0.0252 \text{ Temperature} * \text{KH}_2\text{PO}_4 - 0.1703 \text{ Time} * \text{PHB} + 0.0006 \text{ Time}$$
$$* \text{K}_2\text{HPO}_4 + 0.0844 \text{ Time} * \text{KH}_2\text{PO}_4 - 0.0177 \text{ PHB} * \text{K}_2\text{HPO}_4 + 0.0002 \text{ PHB} * \text{KH}_2\text{PO}_4$$
$$+ 0.0821 \text{ K}_2\text{HPO}_4 * \text{KH}_2\text{PO}_4$$

The contour plot (Fig 6D) is used to predict optimal levels of components for different test variables. Two variables, time and temperature, were used to observe the enzyme activity of PHB depolymerase from *A. caviae* Kuk1-(34) sp. while keeping the hold values of PHB, $KH_2PO_4$, and $K_2HPO_4$ as zero. It was observed from the contour plot that PHB depolymerase from *A. caviae* Kuk1-(34) sp. gave maximum enzyme activity with an increase in the incubation time (9 days) and decrease in the temperature (17 °C).

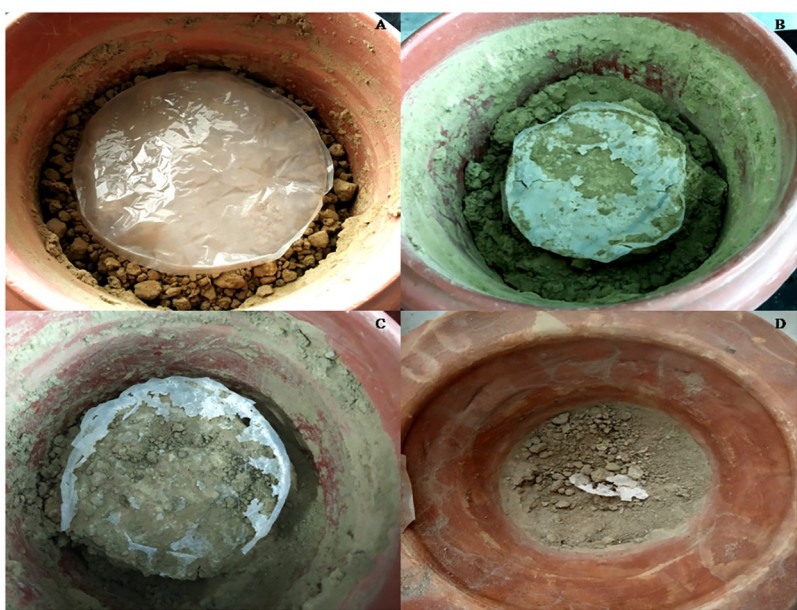

**Fig 7. Soil burial method for the degradation of PHB film as compared with (A) control after treatment with the** *Aeromonas caviae* **Kuk1-(34) sp. for (B) 15 days (C) 30 days and (D) 45 days.**

## 3.9. Soil bioremediation and solid waste management with *A. caviae* Kuk1-(34) sp. under optimized conditions

The optimized parameters obtained from CCD were further used for solid waste management by soil burial method. Autonomous factors like time, $KH_2PO_4$, $K_2HPO_4$ with (+1) values and combined interaction of time and PHB with (-1) values were used for the soil burial method treated with Kuk1-(34) strain. After the incubation for 15, 30, and 45 days, soil-buried PHB polymer film was taken from soil pots. It was observed that maximum degradation occurred when incubation was done for 45 days (Fig 7). The weight loss of PHB film after 45 days of incubation was 94.4% after the soil burial method (Table 12), which is 8.64% more under optimized conditions as compared to the weight loss in unoptimized conditions. This indicates that the CCD model validates the optimized values that can be used on the industrial level for the degradation of PHB based bioplastics.

## 4. Discussion

A rapid increase in the production and utilization of synthetic plastics on a daily basis and their non-degradability in nature leads to the introduction of biodegradable PHB-based bioplastics. A wide variety of micro-organisms accumulate PHB as intracellular granules in a

**Table 12. Weight loss degradability of PHB film after different incubation time in optimised conditions.**

| Bacterial species | Weight of PHB film (g) | Weight of pre-treated soil buried PHB film (g) | | | Weight loss degradability (%) after 45 days |
|---|---|---|---|---|---|
| | | 15 days | 30 days | 45 days | |
| Control | 0.0250 | 0.0250 | 0.0250 | 0.0250 | 0.00% |
| *Aeromonas caviae* Kuk1-(34) sp. | 0.0250 | 0.0198 | 0.0089 | 0.0014 | 94.4% |

highly reduced and insoluble polymer state [24]. In the present study, soil samples were collected from sewage waste from different locations and grown on BHM+PHB medium. BHM contain all nutrients except carbon source, necessary for the growth of bacteria. Only those bacteria that are able to decompose hydrocarbon will grow in this media. Specific carbon source i.e. hydrocarbon (in this case PHB) can be added to this medium and their utilization in terms of hydrolysis can be studied. Out of 127 isolates, a total of 22 were positive PHB degraders; among them only three displayed significant growth and exhibited maximum PHB hydrolysis. The largest zone (11.3 mm) was produced by the bacterial isolate *A. caviae* Kuk1-(34) sp. on BHM media containing 0.15% (w/v) of PHB when incubated for seven days at 37 ºC. This isolate was consequently selected as a prominent PHB depolymerase producer for the degradation PHB based bioplastic. A fungal isolate, *Penicillium citrinum* S2, produced PHB depolymerase when grown in BHM containing 0.2%, w/v PHB [25].

In literature, several articles are available regarding the isolation of different bacterial and fungal species that are potent degraders of PHB-based bioplastics. In one study, Mergaert et al. [26] isolated PHB and P(3HB-co-3HV) copolymer, while Elbanna et al. [27] described *Pseudomonas indica* K2 and *Schlegelella* thermo depolymerase as the PHA degrader. Similarly, Sayyed et al. [28] isolated PHB degrader strain from soil microbes on MSM containing PHB as sole carbon source. Typical enzyme assays for PHB depolymerase have also been described earlier by several researchers. In our study, isolated strain *A. caviae* Kuk1-(34) sp. was analyzed for the PHB depolymerase assay. Previous investigation conducted by Sayyed et al. [22] using *Stenotrophomonas* sp. RZS7 yielded 0.721 U/mL/min of PHB depolymerase after four days of incubation. Likewise, the yield of enzyme production was found to be 0.721 U/mL under unoptimized conditions [23]. However, in our case, the strain *A. caviae* Kuk1-(34) sp. yielded about 2.0623 U/mL/min of PHB depolymerase after incubation for seven days at 37 ºC [22]. Similar enzyme activities for the PHB depolymerase have been reported by many researchers [17, 24, 28, 29]. Up to 85.76% of total weight of PHB film was degraded after the pre-treatment with *A. caviae* Kuk1-(34) sp. after the incubation for 45 days in the soil under unoptimized conditions (Table 5), while no changes were observed in control PHB film (without pre-treatment). This data is quite significant than those of Bano et al. [17] regarding the degradation of PHB film using *Paenibacillus alvei* PHB28. As reported by Pati et al. [30], *Bacillus* sp. C1 (KF626477) showed significant PHB degradation within 7–21 days. Similarly, 87.74% biodegradation of PHB was obtained using isolate *Stenotrophomonas* sp. RZS7 under natural soil environment [22].

SEM analysis of the PHB polymer film was used to observe the progress of degradation. In the present study, surface analysis of treated PHB polymer film after incubation of 15, 30, and 45 days in soil burial method showed significant variation in the morphology of PHB polymer film. There was clear visualization of cracks and holes on the surface of the PHB polymer film. Morphological analysis of polyethylene surface comparing with control film was done by SEM and was reported by Gautam and Kaur [31]. Another study conducted by Calabia and Tokiwa [32] observed the growth of *Streptomyces* sp. SC-17 on the surface of the PHB film, which was responsible for the presence of crust and holes [22, 32]. Microbial degradation from soil burial method of the PHB film was also reported by Wen and Lu [33]. A similar study reported 15% of PHB film degradation in the soil after 45 days of incubation and observed surface morphology under SEM [17].

Treated PHB films were also analyzed by FTIR. The findings suggest that the chemical structure of PHB possesses molecules terminated by a hydroxyl and a carboxyl group. The hydroxyl and carboxyl end functional groups showed peaks at approximately 3435 cm$^{-1}$ and 1729 cm$^{-1}$, respectively. Previous characteristics of PHB vibrations were found to be around 1290 cm$^{-1}$ and 980 cm$^{-1}$. The peak at 1290 cm$^{-1}$ determines the–C-O-C- group, while the peak

at 980 cm$^{-1}$ can be allocated to bending vibrations of olefinic -C-H [34]. As deprivation begins with time, it incorporates vinyl (crotonate) ester, and carboxyl groups end groups in PHB structure [35].

Consequently, a steady rise in crotonate ester groups with extrusions paths can be expected, as well as a decline in hydroxyl groups available in the original polymer. The absorption band assigned to stretching vibrations of double carbon/carbon bond, -C = C-, is to be shown at around between 1600 and 1700 cm$^{-1}$ [36]. Band at 1729 cm$^{-1}$, allocated originally to the carbonyl absorption band in infrared spectra, is shifted to 1633 cm$^{-1}$ when combined with vinyl end groups after 45 days [37]. The availability of absorption bands associated with the formation of new chemical groups due to deprivation mechanisms of the polymer was noticed at 2923.08 cm$^{-1}$ and 1047.18 cm$^{-1}$, respectively.

Traditionally, improving one parameter at a time is exhaustive and expensive; therefore, statistical methods are utilized for optimization [38]. Plackett and Burman's statistical method involves a two-level fractional factorial saturated strategy that uses only treatment combinations to estimate the main effects of factors independently, assuming that all interactions are insignificant [39]. Full factorials design the number of factors increases exponentially leading to an unmanageable number of experiments [39]. Hence, fractional factorial design like Plackett-Burman becomes a method of choice for initial screening of medium components [23].

To maximize enzyme output yield, PB design and RSM were implemented, which was demonstrated to be an effective screening method for significant medium components and their optimum amounts for total yield. In this study, to increase the enzyme production by *A. caviae* Kuk1-(34) sp., four factors were screened out from 10 factors by PB design. These four factors were further optimized for the RSM using CCD, where maximum enzyme activity of *A. Caviae* Kuk1-(34) sp. was obtained at 14.98 U/min/mL ($p$<0.05). Shivkumar [40] reported the production of depolymerase from *Penicillium expansum* using Placket Burman design. Bansal et al. [41] reported *Aeromonas punctate* sp. for the production and optimization of depolymerase enzyme using PB design and RSM [41]. Similarly, the production of PHB depolymerase from *E. minima* W2 (PhaZ$_{Emi}$) was studied [42]. The study revealed the importance of carbon sources in growth medium for the production of depolymerase enzymes, as the rate of polymer degradation was affected by the source of carbon [41, 43]. Findings from our CCD model suggest that autonomous factors like $K_2HPO_4$, time, $KH_2PO_4$, and interactive effect of time with PHB were the crucial variables in the PHB depolymerase enzyme production [23].

To validate the above study, a soil burial method with significant factors screened from RSM was applied on the PHB film to observe the degradation of PHB polymer film. In the present study, it is observed that the degradation rate of PHB polymer film increased from 85.76% to 94.4% after PHB film was treated with *A. caviae* Kuk1-(34) sp. in optimized conditions. This remarkable degrading characteristic of the strain was utilized for the soil bioremediation to deal with solid waste management. Since degradation proceeds at natural environmental conditions of temperature (35–37 °C), pH 7.0, presence of nutrients in the soil, and oxygen availability allow faster microbial growth by utilizing PHB film as carbon source. So, the soil was the most promising ecosystem for PHB degradation in the presence of microbial activity that is enzymatic degradation is more relevant than composting or chemical treatment for solid waste.

## 5. Conclusion

The isolated strain *A. caviae* Kuk1-(34) sp. from sewage waste showed great potential to degrade PHB-based biodegradable polymer film almost completely in soil under optimized conditions within a limited time period. The growth and the maximum enzyme activity of the

PHB depolymerase enzyme assay of the isolated strain were observed at ambient temperature (27˚C), indicating that the isolate has the capability to survive well under natural environmental conditions. SEM and FTIR analysis of the PHB polymer film confirmed morphological and structural changes due to the biodegradation of polymer film. The statistical analysis for production optimization recognized significant factors of the medium for increased production of PHB depolymerase enzyme that saves time and currency as well. It can be concluded that *A. caviae* Kuk1-(34). isolated from sewage waste emerges as a potent producer of extracellular PHB depolymerase enzyme, having the potential to act as a bio-catalyst for biodegradation of PHB-based bioplastics for large scale bioremediation of soil.

## Supporting information

**S1 Data.**
(XLSX)

## Acknowledgments

The authors are very thankful to the Deanship of Scientific Research at the Majmaah University for providing the facilities.

## Author Contributions

**Conceptualization:** Qamar Zia, Mohd. Rehan Zaheer,  Roohi.

**Data curation:** Mohammad Amir, Naushin Bano, Abu Baker, Mohammad Shariq, Md Sarfaraz Nawaz.

**Formal analysis:** Mohammad Amir, Naushin Bano, Abu Baker, Qamar Zia, Mohd. Farhan Khan,  Roohi.

**Funding acquisition:** Qamar Zia, Saeed Banawas,  Roohi.

**Investigation:** Mohammad Amir, Abu Baker.

**Methodology:** Mohammad Amir, Naushin Bano, Abu Baker, Mohd. Rehan Zaheer, Mohammad Shariq, Mohd. Farhan Khan,  Roohi.

**Project administration:** Saeed Banawas, Z. R. Azaz Ahmad Azad,  Roohi.

**Resources:** Mohd. Rehan Zaheer, Mohammad Shariq, Md Sarfaraz Nawaz, Z. R. Azaz Ahmad Azad,  Roohi.

**Software:** Mohammad Shariq, Md Sarfaraz Nawaz.

**Supervision:** Qamar Zia, Mohd. Rehan Zaheer,  Roohi.

**Validation:** Naushin Bano, Qamar Zia, Saeed Banawas, Z. R. Azaz Ahmad Azad.

**Visualization:**  Roohi.

**Writing – original draft:** Mohammad Amir, Qamar Zia, Saeed Banawas,  Roohi.

**Writing – review & editing:** Mohammad Amir, Naushin Bano, Abu Baker, Qamar Zia, Saeed Banawas, Anamika Gupta, Danish Iqbal,  Roohi.

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
