## [Decision Letter · Decision Letter 0]

20 Sep 2021

PONE-D-21-27980Isolation and optimization of extracellular PHB depolymerase producer Aeromonas caviae Kuk1-(34) for sustainable solid waste management of biodegradable polymersPLOS ONE

Dear Dr. NA,

Thank you for submitting your manuscript to PLOS ONE. After careful consideration, we feel that it has merit but does not fully meet PLOS ONE’s publication criteria as it currently stands. Therefore, we invite you to submit a revised version of the manuscript that addresses the points raised during the review process.

We are forwarding the opinions of two reviewers, so we ask that the authors read it carefully and take steps to correct the manuscript. As noted by one of the reviewers, we suggest a new English review of the manuscript.

We look forward to receiving your revised manuscript.

Kind regards,

Marcos Pileggi, Ph.D

Academic Editor

PLOS ONE

Journal Requirements:

2. Please provide in your methods section the geographical coordinates of the sites from which sewage samples were collected.

7. Please include your tables as part of your main manuscript and remove the individual files. Please note that supplementary tables (should remain/ be uploaded) as separate "supporting information" files

Reviewers' comments:

Reviewer's Responses to Questions

**Comments to the Author**

1. Is the manuscript technically sound, and do the data support the conclusions?

Reviewer #1: Yes

Reviewer #2: Yes

2. Has the statistical analysis been performed appropriately and rigorously? 

Reviewer #1: Yes

Reviewer #2: Yes

3. Have the authors made all data underlying the findings in their manuscript fully available?

Reviewer #1: Yes

Reviewer #2: Yes

4. Is the manuscript presented in an intelligible fashion and written in standard English?

Reviewer #1: Yes

Reviewer #2: Yes

5. Review Comments to the Author

Reviewer #1: 1. Authors did an excellent job of explaining the concept clearly and accurately.

2. Author should be congratulated for the novel endeavour, however language throughout the MS has to be improved, typographical errors, grammatical errors and syntax errors should be removed before submission.

3. Uniformity should be maintained, while writing the author's name.

4. There is a variation of statement between line no. 43 & 52. Author should check that properly.

5. In line no. 89, 'soil waste' should be replaced by 'sewage waste soil'.

6. In line no. 93, author should only mention the abbreviated form of 'PHB', as the full form is already mentioned in the introduction section.

7. Recast line no. 140.

8. The sub-section 2.11. should be deleted as this protocol is already mentioned in the sub-section 2.7. Author should properly check and modify that.

9. Sub-section 3.1. should be merged with 3.2. and in line no. 198 'soil sample' should be replaced by 'sewage waste soil sample'.

10. In line no. 201, before screening how did they get confirmed that all the 22 isolates were PHB degraders? Author should rearrange the result.

11. In Fig. 1, how the author measure zone of diameter by continuous streaking?

12. In line no. 141, it is already mentioned that 0.0245 g PHB film was used for soil burial, then why term 'different' is used in line no. 222.

13. If only A. caviae Kukl-(34) strain was examined for PHB film degradation, then why did author stated that different strains were examined in line no. 223?

14. Author should properly explain the requirement of section 4. This section includes the results, which were already stated in the individual section.

15. Result pertaining to percentage of PHB degradation should be mentioned in sub-section 3.5. instead of section 4, line no. 322.

16. The total weight of PHB degraded after pre-treatment with A. caviae Kukl-(34) varied in line no. 43, 52, 322 & 349. Author should check it properly.

17. In line no. 352, insert space between 'Bano' and 'et al'.

18. All the tables should be arranged as per the sequence of result throughout the MS. For eg. Table 2 should be captioned as Table 1 as it is the first table in the result section.

19. Reports are available (Pati et al., 2020, https://doi.org/10.1007/s00284-020-01922-7) for PHB degradation within 7-21 days by composting method. Thus, author should compare degradation of PHB in laboratory as well as environmental condition.

20. Throughout the MS, term 'the' is used repeatedly. Author should look into that and rectify wherever necessary.

The MS is accepted for publication; however, minor modification is required as per suggestion.

Reviewer #2: 1) Line 40- depolymerase enzyme activity – change it to depolymerase activity

2) Line 44 - FTIR, SEM- Expand

3) Line 48 - ANOVA

4) Line 51- This strain – this culture

5) Line 52 – was found to be- rephrase

6) 2.2. Sample collected- Provide coordinates/GPS location of the sampling site

7) Line 100- hours – h

8) Line 122- Cite the reference for Bergey's Manual of Determinative Bacteriology

9) Line 123- Table 3 ?

10) Line 191 – isolated bacterial isolate?

11) Line 227- SEM analysis – what is SEM?

12) In Methodology – details of SEM are not mentioned

13) Line 307-308 - 4. Application of A. caviae Kuk1-(34) sp. for soil bioremediation by soil burial method in aspects of solid waste management. What is author intend by in aspect of solid….?

14) Discussion can be improved

6. PLOS authors have the option to publish the peer review history of their article (what does this mean?). If published, this will include your full peer review and any attached files.

Reviewer #1: **Yes: **Dr. Deviprasad Samantaray

Reviewer #2: **Yes: **R. Z. Sayyed

---

## [Author Response · Author response to Decision Letter 0]

29 Jan 2022

Comment 1. Please provide in your methods section the geographical coordinates of the sites from which sewage samples were collected.

Compliance: The geographical coordinates of the sites are illustrated in the Table 1 and now mentioned in the section 2.2.

Comment 2. We suggest you thoroughly copyedit your manuscript for language usage, spelling, and grammar. If you do not know anyone who can help you do this, you may wish to consider employing a professional scientific editing service.

Compliance: As per the suggestion, we thoroughly edit the manuscript for language usage, spelling, and grammar.

Comment 3. We note that the grant information you provided in the ‘Funding Information’ and ‘Financial Disclosure’ sections do not match. 

Compliance: It is now corrected.

Comment 4. In your Data Availability statement, you have not specified where the minimal data set underlying the results described in your manuscript can be found. PLOS defines a study's minimal data set as the underlying data used to reach the conclusions drawn in the manuscript and any additional data required to replicate the reported study findings in their entirety. All PLOS journals require that the minimal data set be made fully available. For more information about our data policy, please see http://journals.plos.org/plosone/s/data-availability.

Compliance: We have submitted some raw data.

Comment 5. PLOS requires an ORCID iD for the corresponding author in Editorial Manager on papers submitted after December 6th, 2016. Please ensure that you have an ORCID iD and that it is validated in Editorial Manager. To do this, go to ‘Update my Information’ (in the upper left-hand corner of the main menu), and click on the Fetch/Validate link next to the ORCID field. This will take you to the ORCID site and allow you to create a new iD or authenticate a pre-existing ID in Editorial Manager. Please see the following video for instructions on linking an ORCID iD to your Editorial Manager account: https://www.youtube.com/watch?v=_xcclfuvtxQ

Compliance: ORCID iD of the corresponding author is mentioned in the revised MS and also will be mentioned while resubmission in the portal.

Comment 6. Please include your tables as part of your main manuscript and remove the individual files. Please note that supplementary tables (should remain/ be uploaded) as separate "supporting information" files

Compliance: All tables are now included in the revised MS after the references and figure caption.

Point-by-point responses to the Reviewer #1 comments

Manuscript Number: PONE-D-21-27980

Title: Isolation and optimization of extracellular PHB depolymerase producer Aeromonas caviae Kuk1-(34) for sustainable solid waste management of biodegradable polymers

We are grateful to the reviewer for his careful reading of our manuscript, and for providing useful feedback to clarify both of our results and conclusions. The concerns raised by the Reviewer have all been addressed and highlighted with Turquoise colour which is summarized below.

Comment 1. Authors did an excellent job of explaining the concept clearly and accurately.

Compliance: Authors are very thankful to the reviewer for his appreciation. 

Comment 2. Author should be congratulated for the novel endeavour, however language throughout the MS has to be improved, typographical errors, grammatical errors and syntax errors should be removed before submission.

Compliance: Authors are very thankful to the reviewer for his appreciation. However as per the suggestion, typographical errors, grammatical errors and syntax errors have been checked and removed.

Comment 3. Uniformity should be maintained, while writing the author's name. 

Compliance: Uniformity has been kept.

Comment 4. There is a variation of statement between line no. 43 & 52. Author should check that properly.

Compliance: Line 43 (now line no. 50) indicates result of soil burial method without optimized condition and line 52 (now line no. 61) indicates result of soil burial method using optimized condition from CCD.

Comment 5. In line no. 89, 'soil waste' should be replaced by 'sewage waste soil'.

Compliance: As per the suggestion of the reviewer, “soil waste” is replaced with “sewage waste soil”. 

Comment 6. In line no. 93, author should only mention the abbreviated form of 'PHB', as the full form is already mentioned in the introduction section.

Compliance: As per the suggestion the abbreviated form of PHB has been added. 

Comment 7. Recast line no. 140.

Compliance: As per the suggestion, line no. 140 has been recast. 

Comment 8. The sub-section 2.11. should be deleted as this protocol is already mentioned in the sub-section 2.7. Author should properly check and modify that.

Compliance: As per the suggestions, section 2.11 was checked and modified. This section explains the methodology of soil burial method using statistically optimized parameters obtained from CCD and is clearly distinct form section 2.7 that deals with soil burial method without optimized conditions. Moreover, we have modified the title of this section to make it more understandable.

Comment 9. Sub-section 3.1. Should be merged with 3.2. and in line no. 198 'soil sample' should be replaced by 'sewage waste soil sample'

Compliance: As per the suggestion, sub-section 3.1 is merged with 3.2 and the word “soil sample” is replaced with “sewage waste soil samples”.

Comment 10. In line no. 201, before screening how did they get confirmed that all the 22 isolates were PHB degraders? Author should rearrange the result.

Compliance: After the serial dilution the sewage waste soil samples was spread on BHM+PHB agar plates. The positive colonies form zone of hydrolysis as shown in Fig 1 on BHM+PHB agar plates. These colonies were marked and selected for further experiments. Also mentioned in Results section.

Comment 11. In Fig. 1, how the author measure zone of diameter by continuous streaking?

Compliance: Fig 1A and 1B were used to measure the zone of diameter as the circular colonies were there and continuous streaking in Fig 1C, 1D and 1E is just to show the zone of hydrolysis. 

Comment 12. In line no. 141, it is already mentioned that 0.0245 g PHB film was used for soil burial, then why term 'different' is used in line no. 222.

Compliance: Since the PHB films were prepared in the laboratory using PHB powder (Sigma), so there was definitely a variation in the weight of the PHB film used for the soil burial method. Therefore, in the manuscript and in the Table 2A, we have mentioned the range of the weight of the PHB film (0.024 – 0.026 g) that was used in the soil burial method. In order to maintain similarity and avoid confusion, we have used the same weight range in methodology and results section.

Comment 13. If only A. caviae Kukl-(34) strain was examined for PHB film degradation, then why did author stated that different strains were examined in line no. 223?

Compliance: It is clearly mentioned in Table 2A that along with the strain Aeromonas caviae Kuk1-(34), two different strain i.e., CA6-(55) and CB2-(20) were also examined for the PHB film degradation in soil. Therefore we stated different strains were examined.

Comment 14. Author should properly explain the requirement of section 4. This section includes the results, which were already stated in the individual section.

Compliance: In the section 4 (now section 3.9), it explains the application of the bacterial strain Aeromonas caviae Kuk1-(34) for soil burial method in totally optimized conditions. The new result section 3.4 explains the soil burial method for unoptimized conditions. It has been found that weight loss of PHB film is 8.64% more in optimized conditions as compared to the weight loss of PHB film in unoptimized conditions. 

Comment 15. Result pertaining to percentage of PHB degradation should be mentioned in sub-section 3.5. instead of section 4, line no. 322.

Compliance: Section 3.5 (now section 3.4) indicates the result of weight loss analysis of PHB film by soil burial method under the un-optimized conditions whereas in section 4 (now section 3.9), line no. 306, the result 94.4% indicates the degradation of PHB film using the optimized conditions from CCD. An increment of 8.64% was found during the weight loss of PHB film under optimized conditions.

Comment 16. The total weight of PHB degraded after pre-treatment with A. caviae Kukl-(34) varied in line no. 43, 52, 322 & 349. Author should check it properly.

Compliance: Line no. 43 (now line no. 50) indicates the result of PHB film degradation without using the optimized condition whereas line no. 52 (now line no. 61) and line 322 (now line 400) indicates the result of PHB film degradation using the optimized conditions after the CCD. As per the suggestion, line no. 349 (Line 340) is checked and corrected to 85.76% as this result also indicates the PHB film degradation without using the optimized conditions.

Comment 17. In line no. 352, insert space between 'Bano' and 'et al'.

Compliance: As per the suggestion, space is inserted between Bano and et al.

Comment 18. All the tables should be arranged as per the sequence of result throughout the MS. For eg. Table 2 should be captioned as Table 1 as it is the first table in the result section.

Compliance: As per the suggestion, table are checked and rearranged. 

Comment 19. Reports are available (Pati et al., 2020, https://doi.org/10.1007/s00284-020-01922-7) for PHB degradation within 7-21 days by composting method. Thus, author should compare degradation of PHB in laboratory as well as environmental condition.

Compliance: As per the suggestion, we have mentioned the reference for Pati et al. (2020) in the discussion section as a comparative study of our work. 

Comment 20. Throughout the MS, term 'the' is used repeatedly. Author should look into that and rectify wherever necessary.

Compliance: As per the suggestion, unnecessary use of term “the” have been checked and rectified.

Point-by-point responses to the Reviewer #2 comments

We are grateful to the reviewer for his careful reading of our manuscript, and for providing useful feedback to clarify both of our results and conclusions. The concerns raised by the reviewer have all been addressed and highlighted with yellow color which is summarized below. 

Comment 1. Line 40- depolymerase enzyme activity – change it to depolymerase activity

Compliance: As per the suggestion depolymerase enzyme activity has been replaced with depolymerase activity

Comment 2. Line 44 - FTIR, SEM- Expand

Compliance: As per the suggestion, FTIR and SEM have been expanded.

Comment 3. Line 48 – ANOVA

Compliance: As per the suggestion, ANOVA has been expanded.

Comment 4. Line 51- This strain – this culture

Compliance: As per the suggestion “this strain” word has been replaced with “this culture” word.

Comment 5. Line 52 – was found to be- rephrase

Compliance: As per the suggestion “was found to be” rephrased in the manuscript.

Comment 6. 2.2. Sample collected- Provide coordinates/GPS location of the sampling site

Compliance: As per the suggestion, the coordinates of the sampling site are mentioned in the Table 1, section 2.2.

Comment 7. Line 100- hours – h

Compliance: As per the suggestion, modification hours into “h” have been done (Line 108).

Comment 8. Line 122- Cite the reference for Bergey's Manual of Determinative Bacteriology

Compliance: As per the suggestion, the citation of Bergey`s Manual of Determinative Bacteriology has been added (Bergey, Krieg and Holt, 1984).

Comment 9. Line 123- Table 3?

Compliance: As per the suggestion, Tables have been checked and renumbered.

Comment 10. Line 191 – isolated bacterial isolate?

Compliance: As per the suggestion, this line has been checked and removed.

Comment 11. Line 227- SEM analysis – what is SEM?

Compliance: As per the suggestion, SEM has been expanded

Comment 12. In Methodology – details of SEM are not mentioned

Compliance: In methodology section 2.8, details of SEM are explained from line no. 153-57.

Comment 13. Line 307-308 - 4. Application of A. caviae Kuk1-(34) sp. for soil bioremediation by soil burial method in aspects of solid waste management. What is author intend by in aspect of solid….?

Compliance: In section 4 (Noe section 3.9), regarding application, we are actually focusing on the degradation of PHB based bioplastics which are dumped in the soil as a solid structure. Therefore, we have mentioned the term solid waste management. Nevertheless, we changed the subtitle to accommodate the reviewer’s comment.

Comment 14. Discussion can be improved

Compliance: As per the suggestion, discussion has been modified.

---

## [Decision Letter · Decision Letter 1]

7 Feb 2022

Isolation and optimization of extracellular PHB depolymerase producer Aeromonas caviae Kuk1-(34) for sustainable solid waste management of biodegradable polymers

PONE-D-21-27980R1

Dear Dr. NA,

We’re pleased to inform you that your manuscript has been judged scientifically suitable for publication and will be formally accepted for publication once it meets all outstanding technical requirements.

Kind regards,

Marcos Pileggi, Ph.D

Academic Editor

PLOS ONE

Additional Editor Comments (optional):

Reviewers' comments:

Reviewer's Responses to Questions

**Comments to the Author**

1. If the authors have adequately addressed your comments raised in a previous round of review and you feel that this manuscript is now acceptable for publication, you may indicate that here to bypass the “Comments to the Author” section, enter your conflict of interest statement in the “Confidential to Editor” section, and submit your "Accept" recommendation.

Reviewer #1: All comments have been addressed

Reviewer #2: All comments have been addressed

2. Is the manuscript technically sound, and do the data support the conclusions?

Reviewer #1: Yes

Reviewer #2: Yes

3. Has the statistical analysis been performed appropriately and rigorously? 

Reviewer #1: Yes

Reviewer #2: Yes

4. Have the authors made all data underlying the findings in their manuscript fully available?

Reviewer #1: Yes

Reviewer #2: Yes

5. Is the manuscript presented in an intelligible fashion and written in standard English?

Reviewer #1: Yes

Reviewer #2: Yes

6. Review Comments to the Author

Reviewer #1: Authors should be congratulated for doing an excellent job to improve the quality of research article by making desirable modifications.

Reviewer #2: The authors have addressed all the concerns and made the corrections as per my suggestion. The manuscript is greatly improved

7. PLOS authors have the option to publish the peer review history of their article (what does this mean?). If published, this will include your full peer review and any attached files.

Reviewer #1: **Yes: **Dr. Deviprasad Samantaray

Reviewer #2: **Yes: **RIYAZALI ZAFARALI SAYYED

---

## [Editor Report · Acceptance letter]

6 Apr 2022

PONE-D-21-27980R1 

Isolation and optimization of extracellular PHB depolymerase producer *Aeromonas caviae* Kuk1-(34) for sustainable solid waste management of biodegradable polymers 

Dear Dr. Roohi:

I'm pleased to inform you that your manuscript has been deemed suitable for publication in PLOS ONE. Congratulations! Your manuscript is now with our production department. 

Kind regards, 

on behalf of

Dr. Marcos Pileggi 

Academic Editor

PLOS ONE